# Comparative Analysis of HPV16 Variants in the Untranslated Regulatory Region, L1, and E6 Genes among Vaccinated and Unvaccinated Young Women: Assessing Vaccine Efficacy and Viral Diversity

**DOI:** 10.3390/v16091381

**Published:** 2024-08-29

**Authors:** Kahren van Eer, Tsira Dzebisasjvili, Renske D. M. Steenbergen, Audrey J. King

**Affiliations:** 1National Institute for Public Health and the Environment, Centre for Infectious Disease Control, 3721MA Bilthoven, The Netherlands; kahren.van.eer@rivm.nl (K.v.E.); tsira.dzebisasjvili@rivm.nl (T.D.); 2Department of Pathology, Amsterdam UMC Location Vrije Universiteit Amsterdam, 1007MB Amsterdam, The Netherlands; r.steenbergen@amsterdamumc.nl; 3Cancer Center Amsterdam, Imaging and Biomarkers, 1007MB Amsterdam, The Netherlands

**Keywords:** human papillomavirus 16, vaccination, variants

## Abstract

HPV16 is occasionally detected in vaccinated women who received the bivalent HPV16/18 vaccine, usually at low viral loads. This study explored potential differences in HPV16 variants between vaccinated and unvaccinated women. HPV16-postive viral loads were detected in 1.9% (17/875) and 13% (162/760) of vaccinated and unvaccinated women, respectively, showcasing the vaccine’s high efficacy. The L1, E6, and URR regions of HPV16 were sequenced from genital swabs from 16 vaccinated and 25 unvaccinated women in the HAVANA (HPV Among Vaccinated And Non-vaccinated Adolescents) study. The majority of HPV16 variants from vaccinated and unvaccinated women clustered similarly with sub-lineages A1 and A2. Additionally, a separate cluster within lineage A was found, with the variants sharing the L1-located SNP A753G (synonymous) and the URR-located SNP T340C, which did not occur in the other variants. Furthermore, four variants from vaccinated women had relatively long branches, but were not characterized by specific SNPs. The frequency of G712A in the URR was the only SNP observed to be marginally higher among vaccinated women than unvaccinated women. Non-synonymous SNPs T266A in the FG-loop of L1 and L83V in E6 were common among variants from vaccinated and unvaccinated women, but present in similar frequencies. In conclusion, the detection of HPV16 in vaccinated (and unvaccinated) women seemed to be the result of random circulation within this study population.

## 1. Introduction

Cervical cancer (CC) is the fourth most common cancer among women. In 2020, it was estimated that there were 604,127 cases of CC and 341,831 CC-related deaths globally, with incidence and mortality rates of 13.3 and 7.2 per 100,000 women, respectively. In that same year, the World Health Organization (WHO) launched the “Cervical Cancer Elimination Initiative”, which aims to reduce the burden of CC below the threshold of -four cases per 100,000 women-years in every country by 2030 [1].

Persistent infections with high-risk (hr) human papillomavirus (HPV) types are recognized as the necessary cause for the development of CC [2]. HPV is a double-stranded DNA virus with a circular genome of ~8000 base pairs, with early genes (E1–E7) and late genes (L1–L2), in addition to an upstream regulatory region (URR) [3]. The infection starts in the basal layer of the squamous epithelium, with low-level viral DNA replication by E1 and E2 proteins. As cells differentiate into the suprabasal layer, replication intensifies, and virions are released through the assembly of the L1 and L2 major and minor capsid proteins. The HPV genome may integrate into the host genome, often disrupting the E2 gene. This disruption activates the E6 and E7 proteins, driving viral replication and potentially leading to cancer development [3].

To date, over 200 HPV types have been identified, 20 of which are significantly more prevalent in cancer cases than in women with normal cervical cytology [4]. The most oncogenic types are HPV16 and HPV18, which are associated with 60% and 11% of CC cases, respectively [5]. Three HPV vaccines are currently available against two or more types, which are the bivalent HPV16/18 vaccine, the quadrivalent HPV6/11/16/18 vaccine and the nonavalent HPV6/11/16/18/31/33/45/52/58 vaccine. All vaccines incorporate L1-Viral-Like-Proteins (VLPs) derived from the target HPV types [6].

In 2009, the Netherlands implemented the bivalent vaccine, which was routinely offered to 12-year-old girls and as a catch-up campaign for girls born in 1993–1996 (12–16 years old) in three doses at 0, 1 and 6 months [7]. In 2014, the European Medicines Agency (EMA) licensed a two-dose schedule (0 and 6 months) for girls under the age of 15 [8]. Since 2022, the Dutch Health Board is advising two doses to boys and girls from the age of 10 [9].

Multiple studies have reported on the strong and stable vaccine-generated HPV16 and HPV18 antibody responses [10,11,12,13], in addition to high protection against HPV16/18 infections [14,15] and associated precancerous cervical lesions [16,17,18]. However, vaccine efficacy seems to be lower towards incident HPV16/18 infections compared to persistent HPV16/18 infections of 6 months and 12 months [19,20].

The bivalent HPV16/18 vaccine is based on the major capsid L1 proteins of specific HPV16 and HPV18 variants (acc. AAC09292.1 [21]). However, HPV is a collection of genetically versatile isolates, i.e., variants. Generally, HPV variants differing 1–10% or 0.5–1% on a whole-genome basis are classified into lineages and sub-lineages, respectively [22]. For example, HPV16, the most oncogenic type, consists of four lineages and sub-lineages (A1–A4, B1–B4, C1–C4, D1–D4) [23]. Sequencing data of the HPV16 L1 gene, the E6 oncogene and the untranslated regulatory region (URR) show that the sub-lineages express different degrees of persistence and oncogenicity and vary in their geographical distribution [24,25]. Whether the effect of vaccination differs for the HPV16 lineages, sub-lineages and/or specific variants remains poorly understood.

Previous studies examined variation, mostly focusing on L1 antigenic loops, and speculated on potential consequences for vaccine-induced immunogenicity, but also noted the necessity of examining intra-type variation within HPV16 detected among vaccinated and unvaccinated women [26,27,28]. This study examined the genetic variation within the L1, URR and E6 regions of HPV16 among women vaccinated with the bivalent HPV16/18 vaccine and unvaccinated women enrolled in the “HPV Among Vaccinated And Non-vaccinated Adolescents” (HAVANA) study.

## 2. Materials and Methods

### 2.1. Population Characteristics

From 2009 to 2021, self-collected vaginal swabs were provided annually by young women participating in the HAVANA observational cohort study [29]. In brief, 29,162 girls aged 14–16 were invited to participate in the years 2009–2010, of which 6% consented. Prior to vaccination, a baseline swab was collected from the women. The coverage of HPV vaccination in the Netherlands for the catch-up group born between 1993 and 1996 was approximately 50% [30].

In this study, only women who were fully vaccinated with three doses of the bivalent HPV16/18 vaccine and a matched (age and ethnicity) subset of unvaccinated women were included; women who were positive for HPV16 at baseline were excluded. Informed consent was obtained from study participants and (if possible, both) parents, or a legal representative. This study was approved by the Medical Ethics Committee of VUmc Amsterdam (2009/22).

### 2.2. HPV Genotyping and Viral Load Quantification

Total DNA was extracted from 200 µL of swab fluid with the MagNA Pure 96 platform (Roche Diagnostics, Indianapolis, IN, USA) and eluted in 100 µL of elution buffer. A phocine herpesvirus-1 spike was added to each sample as an internal control for DNA isolation. HPV detection and genotyping was performed with the analytically sensitive and widely used SPF10-DEIA-LiPA25 assay, which is able to detect up to 25 HPV types [31].

Both incident (i.e., HPV16 detected at least once during the follow-up period from 2009 to 2021) and persistent (i.e., HPV16 detected during at least two follow-up rounds in the same period) HPV16 infections were included. For persistent HPV16 infections, only the first genotyping positive measurement was included. Women were not excluded based on general HPV status prior to vaccination. However, it is hypothesized that HPV exposure was minimal, due to the relatively young age of the women (14–15 years).

HPV16 viral load (copies/cell) was obtained by measuring the number of genome equivalents with a HPV16-L1-specific qPCR assay and corrected for cellular content with a β-actin qPCR assay, according to van der Weele (2016) [32]. Only HPV16 viral load positive samples (i.e., HPV16 was detected with the qPCR) were included for downstream processes.

### 2.3. DNA Depletion Shotgun Sequencing

Original swabs with a HPV16 Cp value < 30 were subjected to human and bacterial DNA depletion, based on [33,34]. Briefly, 5 µL 1% saponin was added to 200 µL original swab fluid and incubated for 15 min at RT. Thereafter, 20 µL lysozyme/achromopeptidase mix (10 mg/mL) was added and incubated for 1 h at 37 °C while shaking. Next, a 42 µL mix of 10 U/µL omnicleave, 1M Tris-HCl (pH = 8) and 25 mM MgCl_2_ was added and incubated for 1 h at 37 °C while shaking. Lastly, the swab fluids were centrifuged for 10 min at 13,000 rpm, filtered with 0.45 µM Costar^®^ Spin-X^®^ spinning columns (Corning, Corning, NY, USA), after which the centrifugation was repeated. Total DNA isolation was performed on the supernatant with the InGenius (ELITechGroup, Turin, Italy). Library preparation and next-generation sequencing (NGS) were performed with the nextera XT DNA kit and the NextSeq 500/550 v2.5 kit (mid-output, 300 cycles) from Illumina^®^ (Illumina, San Diego, CA, USA).

### 2.4. Targeted Amplification of L1, URR and E6 Regions

Original swabs with a HPV16 Cp value > 30 were subjected to total DNA isolation with the InGenius platform and targeted amplification of the L1, URR and E6 regions. Separate pre-PCR assays amplified the L1 and URR/E6 regions as a whole, after which each amplicon was re-amplified in four and five overlapping fragments, respectively. The pre-PCR and nested-PCR primers and cycling conditions are stated in Appendix A. Amplicon sequencing was performed with NGS as described in the previous paragraph.

### 2.5. Sequence Assembly

For assembly, the L1 region was selected as it is the target of the bivalent vaccine and the URR and E6 regions were selected for high-resolution phylogenetic classification.

Primer sequences were removed from the raw NGS data with the in-house tool AmpliGone (version 1.2.0). Assembly of L1, URR and E6 sequences was performed by the in-house ViroConstrictor pipeline (version 1.1.0) [35]. This pipeline is specifically designed to process amplicon-generated as well as shotgun-generated sequencing data from multiple sequencing platforms, including Illumina.

### 2.6. Phylogenetic Analyses

Sequences with ≥90% coverage and a reading depth of ≥5 reads were included in subsequent analyses and were deposited into the NCBI Genbank Database (www.ncbi.nlm.nih.gov (accessed on 24 July 2023)) under accession numbers OR348819-OR348859. Maximum likelihood analysis was performed with the IQtree webserver (http://iqtree.cibiv.univie.ac.at/ (accessed on 17 January 2023)) [36] on the combined L1, URR and E6 sequences under the best fitting substitution model (K3Pu + F + I). The following reference sequences used for the construction of the phylogenetic tree were retrieved from the NCBI Genbank Database: K02718.1 (A1), AF536179.1 (A2), HQ644236.1 (A3), AF534061.1 (A4), AF536180.1 (B1), KU053910.1 (B2), HQ644298.1 (B3), KU053911.1 (B4), AF472509.1 (C1), HQ644244.1 (C2), KU053921.1 (C3), KU053925.1 (C4), HQ644257.1 (D1), AY686579.1 (D2), AF402678.1 (D3), KU053933.1 (D4). Maximum parsimony trees with the L1/URR/E6, the URR/E6 and the L1 sequences were constructed with Bionumerics version 7.6.3.

### 2.7. Variant Analyses

The number of unique SNPs per consensus sequence was analyzed in R4.2.2. Differences in viral load and SNP load among vaccinated and unvaccinated women were analyzed with a Mann–Whitney U (MWU) test in R.4.2.2. Differences in individual SNP frequencies among vaccinated and unvaccinated women were analyzed with a Fisher exact test in R4.2.2. A *p*-value ≤ 0.05 was considered statistically significant.

## 3. Results

### 3.1. Population Characteristics and HPV16 Infections

In 2009, the HAVANA study comprised 1832 women, of which 875 women (47.8%) were fully vaccinated with three doses and 760 women were unvaccinated (41.5%) (Figure 1A). Additionally, 197 women (10.8%) were excluded from further analyses, as they did not provide a swab for baseline (i.e., before vaccination) HPV testing and/or had no available vaccination status. HPV16 DNA was detected in 26 vaccinated women (3.0%), of which 24 women (2.7%) tested HPV16-negative at baseline. HPV16 DNA was also detected in 115 unvaccinated women (15.1%), of which 110 women (14.5%) tested HPV16-negative at baseline.

A HPV16-positive viral load was measured for 17 vaccinated women (1.9%) and 103 unvaccinated women (13.6%). Median HPV16 viral load among vaccinated and unvaccinated women was compared, including the viral load negative measurements, which were given an arbitrary value of 1 × 10^−5^ copies/cell [37]. It was found that HPV16 detected among vaccinated women had a significantly lower median viral load of 0.00026 (0.00001–0.0052) copies/cell compared to the median HPV16 viral load of 0.8405 (0.0071–23.2981) detected among unvaccinated women (*p* < 0.0001) (Figure 1B).

Due to the relatively low number of HPV16 (viral load) positive samples from vaccinated women compared to unvaccinated women, all 17 HPV16 viral load positive samples and a matched subset of 34 HPV16 viral load positive samples were selected for next-generation Illumina sequencing. Both the vaccinated and matched subset of unvaccinated women were of Dutch ethnicity.

### 3.2. L1–URR–E6 Sequence Quality

The L1, URR and E6 regions lie adjacent to each other in the HPV genome. Therefore, for each sample, a single consensus sequence of 2827 bases was generated, spanning L1 (1518 bases), the URR (853 bases) and E6 (456 bases). Only L1–URR–E6 consensus sequences with a minimal coverage of 90% and a reading depth of at least five reads per base-position were included in subsequent analyses. In total, 41 sequences with these criteria were successfully generated. Despite very low HPV16 viral load measurements, L1–URR–E6 sequences were successfully generated from 16 out of the 17 samples provided by vaccinated women (Figure 2). For unvaccinated women, 25 out of the 34 selected samples were successfully sequenced for the L1–URR–E6 region. For both vaccinated and unvaccinated women with sequenced samples, the time from the beginning of the study to initial HPV16 infection ranged between 3 and 12 study rounds (Appendix A). Notably, samples with a higher viral load did not necessarily result in better coverage, as is seen from the wider spread of sequence coverage (Figure 2).

To check for potential PCR-generated artifacts, both shotgun and amplicon sequencing was performed on a subset of 5 samples. The sequences obtained by the two sequencing methods were shown to be 100% identical for each sample (Appendix A), therefore allowing combined analysis of amplicon-generated and shotgun-generated sequences.

### 3.3. HPV16 Sequence Diversity and Phylogeny

Maximum parsimony analysis showed the phylogenetic diversity between HPV16 variants (Appendix A). The number of unique SNPs (i.e., not shared by variants from vaccinated or unvaccinated women) were analyzed per L1–URR–E6 consensus sequence. In total, 16 (39%) and 6 (14.6%) variants were found to have ≥1 or ≥2 unique SNPs, respectively (Table 1). Among vaccinated women, 6 (37.5%) and 2 (12.5%) variants were found to have ≥1 or ≥2 unique SNPs, respectively. Among unvaccinated women, 10 (40%) and 4 (16%) variants were found to have ≥1 or ≥2 unique SNPs. Based on the L1–URR–E6 consensus sequences, vaccinated and unvaccinated women were found to have similar numbers of HPV16 variants with ≥1 SNP (*p* = 0.8728).

Maximum likelihood analyses, based on the L1–URR–E6 regions, showed that the majority of the HPV16 variants belonged to lineage A, more specifically sub-lineage A1 or A2 (Figure 3). Notably, four variants within the A lineage had relatively long branches, all belonging to vaccinated women. Furthermore, a separate cluster within lineage A was observed consisting of three variants from unvaccinated women and one variant from vaccinated women. Additionally, one and two variants clustering with sub-lineages B1 and D4 were found in vaccinated women, respectively. However, no significant difference in the number of HPV16 variants from vaccinated and unvaccinated women clustering with sub-lineage A1 (*n* = 6 and *n* = 8, respectively) or sub-lineage A2 (*n* = 9 and *n* = 11, respectively) was found (*p* = 0.90).

### 3.4. Variant-Specific SNP Count

Every L1–URR–E6 consensus sequence was analyzed separately for the number of SNPs compared to reference sequence K02718.1, from sub-lineage A1. The variants from sub-lineages B1/D4 from unvaccinated women were found to have the most SNPs per sample, ranging from 5 SNPs in the E6 region to 15 SNPs in the L1 and URR regions (Appendix A). The number of SNPs detected in the variants belonging to lineage A ranged from only one SNP in the L1, URR and E6 regions to nine SNPs in the URR region. The average number of SNPs per sample was found to be similar for the L1 (*p* = 0.14), URR (*p* = 0.60) and E6 (*p* = 0.72) regions when comparing the group of vaccinated to the group of unvaccinated women (Figure 4).

### 3.5. SNP Diversity and Amino-Acid Changes

The types and positions of SNPs were analyzed more closely (Figure 5). A total of 19, 39 and 12 SNPs (green squares) were found on 8, 4 and 15 different positions across the L1, URR and E6 regions among vaccinated women, respectively. A total of 78, 91 and 31 SNPs (black squares) were found on 31, 32 and 8 different positions across the L1, URR and E6 regions among unvaccinated women, respectively. SNPs were completely absent in 3 (18.9%) and 1 (4%) variants from sub-lineage A1 among vaccinated and unvaccinated women. The SNPs A796G in the L1 region, G37T and G365A in the URR region and T247G in the E6 region were frequently found among the variant population. Furthermore, the SNPs A753G in the L1 region and T340C in the URR region, were observed to be solemnly present in the variants from the separate lineage A cluster (highlighted in blue).

Several SNPs were found to be non-synonymous, including the A796G SNP leading to a T266A amino acid change in the FG-loop of L1 and the T247G SNP leading to a L83V amino acid change in E6 (Table 2). However, individual synonymous or non-synonymous SNPs were not found to be present in significantly different frequencies among vaccinated and unvaccinated women (Table 2, Appendix A). One exception was the frequency of G712A in the URR, which was found in three (18.8%) variants from vaccinated women but was absent variants from unvaccinated women (*p* = 0.05) (Appendix A).

## 4. Discussion

The bivalent HPV16/18 vaccine is highly effective in preventing HPV16/18 infections and associated diseases, and cross-protects against certain oncogenic non-vaccine types, most prominently HPV31 and HPV45 [14,15,16,17,38]. Nevertheless, infections with the vaccine types are occasionally detected in vaccinated women, although in low viral loads [39]. In our study, a positive HPV16 viral load was detected in 1.9% (17/875) of vaccinated women and in 13.6% (162/760) of unvaccinated women during the follow-up from 2009 to 2021, showcasing the vaccine’s high efficacy and in agreement with previous research [40]. Here, we investigated potential differences in HPV16 variants detected among vaccinated and unvaccinated women, based on L1, URR and E6 sequencing. To our knowledge, this is one of the first studies examining HPV16 variation within a vaccine-eligible population.

We report a moderate HPV16 sequence diversity, represented in unique SNPs found along the L1, URR and/or E6. The majority of HPV16 variants clustered with sub-lineages A1 and A2. Previous studies, based on whole genome sequences, showed relatively high diversity for HPV16 variants, the majority of which also clustered with sub-lineages A1 and A2 [41,42,43]. This suggests that the other genes that were not sequenced in this study may contain unique SNPs, but that the classification of lineages, in general, does not change. In addition, we found a separate cluster within lineage A, consisting of one variant and three variants from vaccinated and unvaccinated women, respectively. These variants shared two mutations, the synonymous SNP A753G in L1 and T340C in the URR, which did not occur in the other variants within this study. Furthermore, four variants from vaccinated women had relatively long phylogenetic branches. These variants seemed to only contain the four SNPs (A796G in L1, G37T and G365A in the URR, T247G in E6) that were also commonly present in the other variants.

Extensive global mapping of HPV16 variants has shown the worldwide dispersal of “European” sub-lineages A1 and A2 [44]. The ubiquity of these sub-lineages may explain their high prevalence in our study population. In addition, research showed that immunization with the “European” 114K variant produced antibodies against five phylogenetic HPV16 branches [45]. Furthermore, another study reported no lineage-specific vaccine effectiveness for HPV16. However, vaccine effectiveness >80% against HPV31 lineage C was found, but was slightly negative against HPV31 lineages A/B [46]. This suggests that varying cross-protection may be (sub-)lineage-dependent, while protection against vaccine types may be influenced by the presence of specific SNPs. In this study, the number and type of SNPs were not statistically different among unvaccinated and vaccinated women, although the HPV16 sequences in unvaccinated women seemed to be somewhat more phylogenetically diverse albeit not significant. The in-depth comparison of HPV16 sequences from vaccinated and unvaccinated women showed that only the SNP G712A was slightly increased among vaccinated women. G712A is located on one of the four E2 protein binding sites of the p97 promotor in the URR [47]. The mutation may alter E2 binding, enabling a change in E6/E7 transcription and infection. All in all, more research is necessary to accurately determine the effect of specific SNPs within HPV16 on vaccine effectivity.

The L1 protein contains five immuno-dominant loops (BC, DE, EF, FG, HI), which are located on the capsid protein surface and are accessible to neutralizing antibodies. SNPs within these loops can affect antibody production and binding affinity through changes in the T-cell epitope [48,49,50]. In line with our results, previous research found that T266A in the FG loop is most common in variants from lineage A (“European”) [27]. A recent global analysis of HPV16 L1 amino acid mutations found that the mutations H76Y, T176N, T266A, T353P, and L474F—which were also identified in our study—were most frequently detected in South America, followed by Asia, North America, and Europe [51]. Notably, the T266A mutation, which was highly prevalent in our study population, was reported to be most common in North America, followed by Europe, South America, and Asia. These findings suggest that these mutations are widespread globally, with T266A being particularly predominant in the Americas and Europe, potentially explaining its high frequency in our study population [51].

The E6-located L83V SNP in sub-lineages A1 and A2 has been associated with persistence [25]. In this study, no difference in the frequency of T266A, L83V and other non-synonymous SNPs was found among vaccinated and unvaccinated women. This suggests that these mutations do not play a role in the presence of HPV16 among vaccinated women. HPV is a DNA virus and is dependent on the host’s replication machinery, which is characterized by its high fidelity and proof-reading capabilities. In addition, essential core functions of HPV-encoded proteins, such as the viral capsid structure formation of the L1, may lead to mutational constraints [52,53].

Similar to previous research, HPV16 viral load was decreased in vaccinated women compared to unvaccinated women, which may indicate immune restriction and subsequent viral clearance [39]. Alternatively, low viral replication may help to evade immune recognition during follow-up of the women and is only recognized and cleared when replication is increased [23]. Apart from the difference in sample sizes, this restriction of HPV replication, especially within vaccinated women, may explain why the variants among vaccinated women seem less phylogenetically diverse, albeit not significantly. Overall, the similar phylogenetic clustering of variants and frequencies of SNPs within variants among vaccinated and unvaccinated women suggests the circulation of HPV16 within the study population is more random and that its presence among vaccinated women is likely not attributed to specific genetic differences.

This study has several strengths and limitations. HPV16 prevalence was generally low in vaccinated women, which affected the statistical power of this study. The majority of infections among vaccinated women cleared or remained undetected after one year or were detected only up to three follow-up study rounds. In unvaccinated women, on the other hand, we found relatively more persistent HPV16 infections, which were detected up to six follow-up study rounds. This suggests vaccination hampers the persistence of HPV16. Furthermore, it was not possible to distinguish between active HPV16 infections and depositions. The low HPV16 viral load in vaccinated women provided challenges for proper sequencing. Despite this, we were able to successfully sequence the majority of HPV16 infections among vaccinated women. In addition, we showed that amplicon-generated and shotgun-generated sequences could be used for combined analysis. Cornet (2012) was able to distinguish 97.7% of the HPV16 variants (931/953) based on the URR [24]. Nevertheless, during our research, a study by Ou (2021) was published showing that while phylogenetic classification based on the URR is very accurate (94% and 100% for sub-lineages and lineages, respectively), assignment based on E2 has higher accuracy (100% for sub-lineages and lineages) [54]. Other research has also reported on the high E2 genetic variability among HPV16 lineages, which varies across different geographical regions [55,56]. Therefore, E2 may be a potential future candidate for phylogenetic research. Future research might also consider the role of minority variants. Lastly, data on antibody responses may have provided insight on the presence of HPV16 in vaccinated women. Unfortunately, these data were not available. Nevertheless, previous research with women from the HAVANA study showed similar vaccine-induced IgG antibody levels 1 year prior to infection, regardless of whether they had an HPV16 infection or not [11]. This suggests that the women in our study exhibited well-functioning immune responses and that the presence of HPV16 may be due to other factors, possibly of virologic nature.

## 5. Conclusions

Based on the L1, URR and E6 regions, we found a moderate HPV16 sequence diversity among vaccinated women and unvaccinated women. The majority of HPV16 variants clustered with sub-lineages A1 and A2 in similar frequencies among vaccinated and unvaccinated women. A separate cluster consisting of one variant and three variants from vaccinated and unvaccinated women was also found, which shared two SNPs, A753G in L1 and T340C in the URR, which were not found within other variants. Furthermore, four variants from vaccinated women had relatively long branches. No individual SNPs were found to be differently present among vaccinated or unvaccinated women, except for G712A in the URR. This position is part of an E2 protein binding site. In both vaccinated and unvaccinated women, variants commonly carried the non-synonymous SNPs T266A in L1 and L83V, also in similar frequencies. Overall, the similar phylogenetic clustering of variants and frequencies of SNPs within variants among vaccinated and unvaccinated women suggests the circulation of HPV16 in this study population is more random and that its presence among vaccinated women is likely not attributed to specific genetic differences.

## Figures and Tables

**Figure 1 viruses-16-01381-f001:**
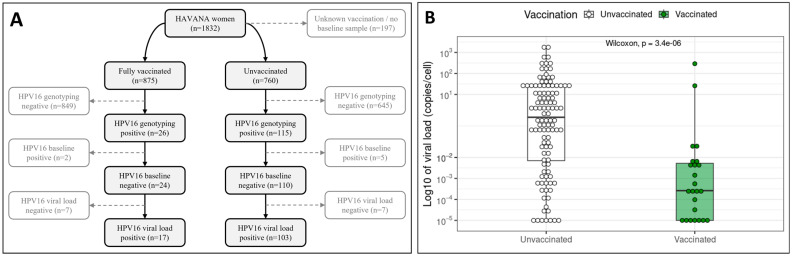
Population characteristics, HPV16 prevalence and HPV16 viral load, with (**A**) an overview of the inclusion numbers in the HAVANA study and the HPV16 prevalence among vaccinated and unvaccinated women; and (**B**) HPV16 viral load measurements among vaccinated (left/white) and unvaccinated (right/green) women.

**Figure 2 viruses-16-01381-f002:**
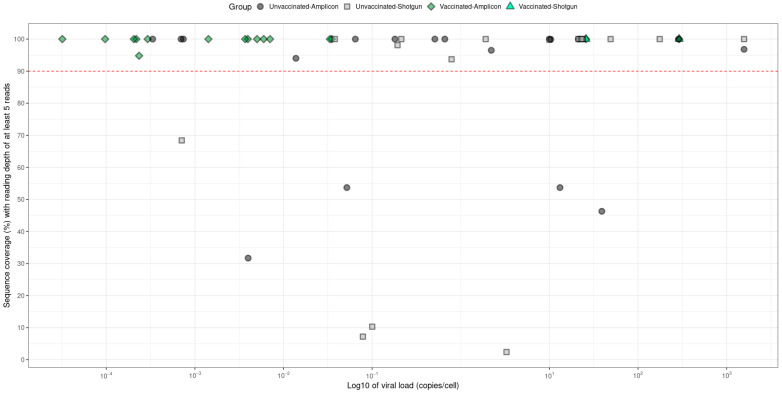
L1–URR–E6 sequence coverage and load per sample. L1–URR–E6 sequence coverage (%) is shown on the Y-axis and the log10 of the HPV16 viral load (copies/cell) on the X-axis. Samples are grouped into Unvaccinated-Amplicon (circle, dark-grey), Unvaccinated-Shotgun (square, light-grey), Vaccinated-Amplicon (diamond, dark-green), Vaccinated-Shotgun (triangle, light-green).

**Figure 3 viruses-16-01381-f003:**
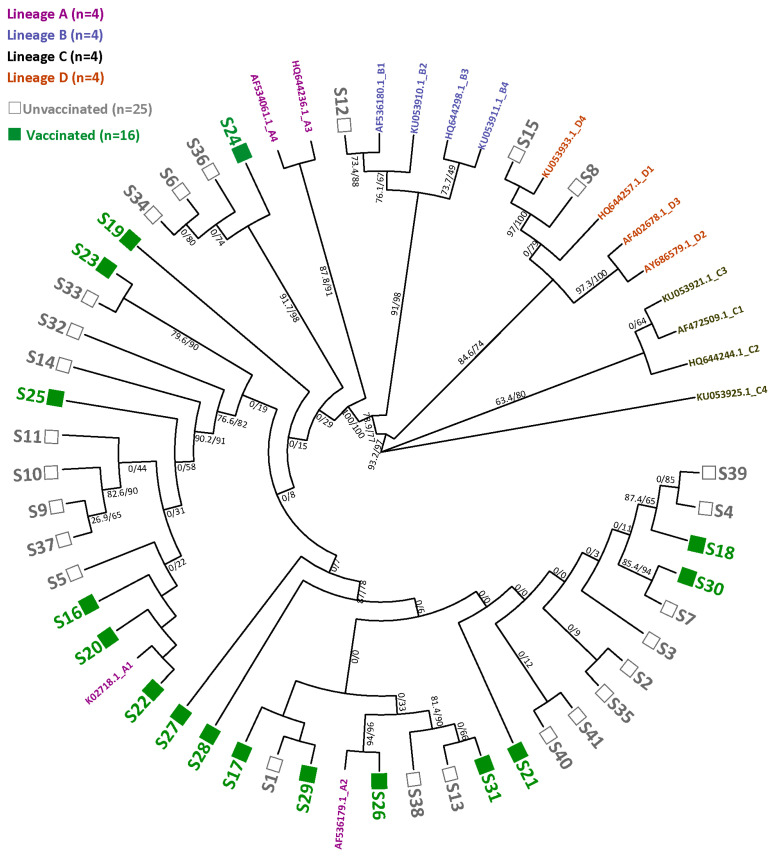
Maximum likelihood tree of the L1–URR–E6 consensus sequences from vaccinated women (green solid squares) and unvaccinated women (grey open squares). Reference sequences belonging to lineages A (violet), B (light purple), C (orange) and D (grey-green) are, respectively, indicated by ‘A1 to A4’, ‘B1 to B4’, ‘C1 to C4’ and ‘D1 to D4’. The long branches belong to S19, S27, S28 and S21, which are all from vaccinated women. The separate cluster is made up of three variants from unvaccinated women (S6, S34, S36) and one variant from vaccinated women (S24).

**Figure 4 viruses-16-01381-f004:**
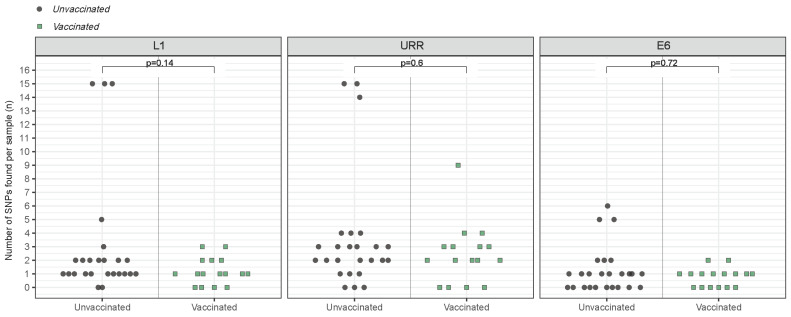
The number of Single Nucleotide Polymorphisms (SNPs) found per HPV16 variant across the L1 (left panel), URR (middle panel) and E6 (right panel) regions in unvaccinated women (grey circles) and vaccinated women (green squares). Sequences were compared with reference K02718.1, from sub-lineage A1.

**Figure 5 viruses-16-01381-f005:**
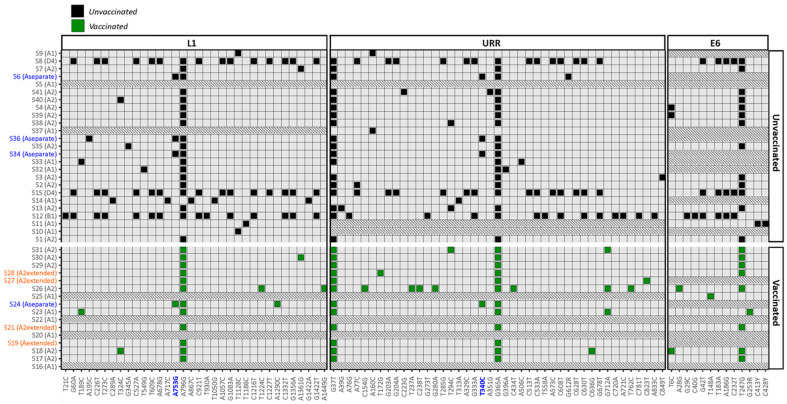
Overview of the position and type of SNP found on each sequence from vaccinated (green squares) and unvaccinated women (black squares) compared to reference sequence K02718.1. Sample names are given on the *y*-axis and position of the SNPs on the *x*-axis. Variants from samples highlighted in blue were found to make up the separate cluster in lineage A. Two SNPs on the *x*-axis, also highlighted in blue, were only found in variants from this separate cluster. The variants from samples highlighted in orange have relatively long branches. Rows which are blocked by diagonal grey stripes represent variants for which no SNPs were found in a specific region.

**Table 1 viruses-16-01381-t001:** The number of HPV16 L1–URR–E6 variants with SNPs that are not shared by other HPV16 L1–URR–E6 variants in the population of vaccinated and unvaccinated women.

	No Unique SNPs	≥1 Unique SNPs	≥2 Unique SNPs
Unvaccinated (*n* = 25)	15 (60)	10 (40)	4 (16)
Vaccinated (*n* = 16)	10 (62.5)	6 (37.5)	2 (12.5)
Total (*n* = 41)	25 (61)	16 (39)	6 (14.6)

**Table 2 viruses-16-01381-t002:** Overview of the non-synonymous SNPs, which give rise to an amino acid change in the L1 and E6 regions. The number and frequency are listed, with the percentage shown in brackets (), of each individual non-synonymous SNP among vaccinated and unvaccinated women; the Fisher *p*-value denotes whether there is a significant difference.

Region	Position	Mutated Triplet	Amino Acid	Vaccinated [*n* = 16, (%)]	Unvaccinated [*n* = 25, (%)]	Fisher *p*-Value
L1	C226T	CAT > TAT	H76Y	0 (0)	3 (12)	0.27
L1 (EF-loop)	C527A	ACC > AAC	T176N	0 (0)	3 (12)	0.27
L1 (FG-loop)	A796G	ACT > GCT	T266A	12 (75)	19 (76)	1
L1 (HI-loop)	A1057G	ACT > CCT	T353P	0 (0)	2 (8)	0.51
L1	T1186C	TCC > CCC	S396P	0 (0)	1 (4)	1
L1	A1290C	AAA > AAC	K430N	1 (6.3)	0 (0)	0.39
L1	A1361G	AAG > AGG	K454R	1 (6.3)	1 (4)	1
L1	T1422G	TTT > TTG	L474F	0 (0)	4 (16)	0.14
E6	A28G	AGA > GGA	R10G	1 (6.3)	0 (0)	0.39
E6	G29C	AGA > ACA	R10T	0 (0)	1 (4)	1
E6	C40G, G42T	CAG > GAT	Q14D	0 (0)	1 (4)	1
E6	G42T	CAG > CAT	Q14H	0 (0)	2 (8)	0.51
E6	T148A	TTA > ATA	L50I	1 (6.3)	0 (0)	0.39
E6	C232T	CAT > TAT	H78Y	0 (0)	3 (12)	0.27
E6	T247G	TTG > GTG	L83V	8 (50)	13 (52)	1

## Data Availability

The original data presented in the study are openly available at GenBank under Accession Numbers OR348819-OR348859.

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
