# Peer review of "Comparative Analysis of HPV16 Variants in the Untranslated Regulatory Region, L1, and E6 Genes among Vaccinated and Unvaccinated Young Women: Assessing Vaccine Efficacy and Viral Diversity"

_viruses, 2024, doi:10.3390/v16091381_

Round 1
Reviewer 1 Report
Comments and Suggestions for Authors
Van Eer et al. have conducted a study to explore differences in HPV16 variants between vaccinated and unvaccinated women as part of the HAVANA study. Despite the strong effectiveness of the bivalent HPV16/18 vaccine, HPV16 was still detected in a small proportion of vaccinated women, specifically 1.9% (17/875), compared to 13.6% (162/760) in unvaccinated women, showcasing the vaccine's high efficacy. The study involved sequencing the HPV16 L1, E6, and URR genes from genital swabs. Most variants from both groups clustered similarly within sub-lineages A1 and A2. A distinct cluster within lineage A featured variants sharing two specific SNPs, but without a clear pattern exclusive to vaccinated individuals. The only significant difference observed was a marginally higher frequency of the SNP G712A in the URR among vaccinated women. Non-synonymous SNPs common to both groups were found with similar frequencies, suggesting that the presence of HPV16 in both cohorts might be due to its random circulation in the population.
The claims are properly placed in the context of the previous literature. The experimental data support the claims. The manuscript is written clearly enough that most of it is understandable to non-specialists. The authors have provided adequate proof for their claims, without overselling them. The authors have treated the previous literature fairly. The paper offers enough details of methodology so that the experiments could be reproduced.
Comments
1. The participants in the HAVANA study were 14-15 years old when vaccinated with the HPV vaccine in 2009-2010, and they underwent annual follow-ups using self-collected vaginal swabs until 2021. This manuscript analyzes the first detected infections of HPV type 16. However, the authors have not reported the interval between the study's start and the first infection. Is it possible to generate a survival plot (1 - survival) to illustrate the time from study inclusion to the first detected HPV type 16 infection for both the vaccinated and unvaccinated groups?
2. Most of the girls in the study tested negative for HPV type 16 at baseline. Did the participants in the HAVANA study report their number of sexual partners prior to inclusion? Additionally, was previous exposure to HPV a criterion for exclusion from the study?
3. In the discussion section, the authors note that 'HPV16 prevalence was generally low in vaccinated women, with the majority of infections either clearing, remaining undetected after one year, or only detected for up to three follow-up years.' Is this observation limited to vaccinated women, or did infections also clear similarly in unvaccinated women? Furthermore, what provisions were in place for women with persistent HPV16 infections over several years? Given that the Netherlands' screening program starts at 30 years of age, some non-vaccinated women in the HAVANA study could potentially experience 10-15 years of persistent HPV16 infection before being detected by the regular cervical cancer screening program.
4. In the HAVANA study, 53.5% (875/1635) of the women were covered by the HPV vaccine. Does this coverage rate reflect the representative uptake for the catch-up group in the Netherlands' HPV vaccination program during its initial years?
5. The HPV 16 viral load detection rate was 13.6% (162/760) in the unvaccinated group during follow-up. Does this rate align with historical data, or is it lower potentially due to herd immunity effects from the HPV vaccine?
Minor revisions
Line 2-4, title, "Comparative Analysis of HPV16 Variants in the Untranslated Regulatory Region, L1, and E6 Genes Among Vaccinated and Unvaccinated Young Women: Assessing Vaccine Efficacy and Viral Diversity"
Line 13-14, abstract, "Despite the strong effectiveness of the bivalent HPV16/18 vaccine, HPV16 is occasionally detected in vaccinated women, usually at low viral loads."
add "In our study, a HPV16 positive viral load was detected in 1.9% (17/875) of vaccinated women, compared to 13.6% (162/760) in unvaccinated women during the follow-up, showcasing the vaccine's high efficacy."
Line 31-33, introduction, "Cervical cancer (CC) is the fourth most common cancer among women. In 2020, it was estimated that there were 604,127 cases of CC and 341,831 CC-related deaths globally, with incidence and mortality rates of 13.3 and 7.2 per 100,000 women, respectively."
Line 37-38, introduction, "Persistent infections with high-risk (hr) human papillomavirus (HPV) types are recognized as the necessary cause for the development of cervical cancer (CC)."
Line 38-39, introduction, "To date, over 200 HPV types have been identified, 20 of which are significantly more prevalent in cancer cases than in women with normal cervical cytology."
Arbyn M, Tommasino M, Depuydt C, Dillner J. Are 20 human papillomavirus types causing cervical cancer? J Pathol. 2014 Dec;234(4):431-5. doi: 10.1002/path.4424. PMID: 25124771.
Line 89-90, materials and methods, "Both incident (i.e., HPV16 detected at least once during the follow-up period from 2009 to 2021) and persistent (i.e., HPV16 detected during at least two follow-up rounds in the same period) HPV16 infections were included."
Line 275-279, discussion, "The bivalent HPV16/18 vaccine is highly effective in preventing HPV16/18 infections and associated diseases, and cross-protects against certain oncogenic non-vaccine types, most prominently HPV31 and HPV45 [12-15; 34]. Nevertheless, infections with the vaccine-types are occasionally detected in vaccinated women, although in low viral loads [35]."
Add: "In our study, a HPV16 positive viral load was detected in 1.9% (17/875) of vaccinated women, compared to 13.6% (162/760) in unvaccinated women during the follow-up from 2009 to 2021, showcasing the vaccine's high efficacy."
Line 355-357, discussion, "Nevertheless, previous research with women from the HAVANA study showed similar vaccine-induced IgG antibody levels 1 year prior to infection, regardless of whether they had an HPV16 infection or not."
Author Response
For the response to reviewer 1, please see the attachment.

Reviewer 2 Report
Comments and Suggestions for Authors
In the manuscript entitled “Variant analysis of the untranslated regulatory region and the L1 and E6 genes of HPV16 detected among vaccinated and unvaccinated young women” the authors present significant data concerning the genetic variability of HPV16 DNA between vaccinated and unvaccinated women. The manuscript is well written and data are sufficiently presented. However, some points need to be addressed.
· The introduction is required to be improved. In particular, authors should provide more details concerning the natural history of HPV infection. Τhe description of viral DNA structure and the molecular mechanisms involved in HPV induced carcinogenesis are required to be mentioned in order to help readers to better understand the biology of HPVs. Moreover, a short description of the design and function of the available HPV vaccines should be added, while the antibody protection that HPV vaccines offer has to be underlined.
· The authors must provide more details about the ethnicity of the examined women.
· A previous large-scale analysis of the publicly available HPV16 L1 gene sequences revealed the global distribution of specific amino acid changes (Viruses. 2022 Dec 31;15(1):141. doi: 10.3390/v15010141). In particular, the T266A mutation (FG loop) was most frequent in North America (34%), followed by Europe (22%), South America (9.1%) and Asia (5.3%). Please further discuss your findings considering the global distribution of HPV16 L1 gene mutations.
· The authors suggest that “E2 may be a potential future candidate for phylogenetic research.” Previous analyses have been performed in different geographic populations, including Europe and it has been suggested that the HPV16 E2 gene can provide significant information about the phylogenetic clustering of viral DNA. Please add the appropriate references.
Author Response
For the response to reviewer 2, please see the attachment.

Round 2
Reviewer 2 Report
Comments and Suggestions for Authors
The manuscript has been sufficiently improved and it is suitable for publication.